# Efficient Deep Reinforcement Learning for Optimal Path Planning

**Jing Ren [1],\*, Xishi Huang [2] and Raymond N. Huang [3]**

1    Department of Electrical, Computer, and Software Engineering, Ontario Tech University,
     Oshawa, ON L1G 0C5, Canada
2    RS OPTO Tech Ltd., Suzhou 215100, China
3    Department of Mechanical and Industrial Engineering, University of Toronto, Toronto, ON M5S 3G8, Canada
*    Correspondence: jing.ren@ontariotechu.ca

**Abstract:** In this paper, we propose a novel deep reinforcement learning (DRL) method for optimal path planning for mobile robots using dynamic programming (DP)-based data collection. The proposed method can overcome the slow learning process and improve training data quality inherently in DRL algorithms. The main idea of our approach is as follows. First, we mapped the dynamic programming method to typical optimal path planning problems for mobile robots, and created a new efficient DP-based method to find an exact, analytical, optimal solution for the path planning problem. Then, we used high-quality training data gathered using the DP method for DRL, which greatly improves training data quality and learning efficiency. Next, we established a two-stage reinforcement learning method where, prior to the DRL, we employed extreme learning machines (ELM) to initialize the parameters of actor and critic neural networks to a near-optimal solution in order to significantly improve the learning performance. Finally, we illustrated our method using some typical path planning tasks. The experimental results show that our DRL method can converge much easier and faster than other methods. The resulting action neural network is able to successfully guide robots from any start position in the environment to the goal position while following the optimal path and avoiding collision with obstacles.

**Keywords:** deep reinforcement learning; global optimal path planning; dynamic programming; mobile robots; shortest path; continuous state space; collision avoidance

## 1. Introduction

Deep reinforcement learning (DRL) has been a powerful tool in many applications, including optimal path planning for mobile robots [1–10]. However, traditional deep reinforcement learning approaches use the trial-and-error method, which is extremely time-consuming and often does not converge [11]. The learning efficiency has become the bottleneck in applying DRL to more real-time path planning problems. One key factor that determines the efficiency of the learning process is the quality of the training data. Traditionally, DRL neural networks learn from randomly generated experience training data. Consequently, training is often a lengthy process and it is easy to get trapped in a local minimum and fail. In order to overcome this fundamental challenge of DRL, in this paper, we propose an improvement to the training data quality by using dynamic programming (DP)-based optimal data collection. This new DP-based data collection method is an excellent match for the path planning problems because the shortest distance path planning problems can be mapped to be a DP problem that can therefore achieve the global optimal solution. This DP-based data collection method can provide an abundant optimal training dataset for DRL.

Path planning is a popular research topic with many recently published studies [12–15]. However, it remains an active research area with much to be explored. In path planning,

researchers strive to achieve the global optimal solution and one such method that can guarantee this is DP. There are many DP algorithms used to solve various problems that can be applied in navigating the global optimal path for mobile robots, which commonly use grid discretization to approximate the global optimal solution [16–19]. Different from most works, in this study, we first mapped the dynamic programming method to the typical, shortest traveling distance path planning for mobile robots and then used DP to find the exact, analytical, optimal solution for the continuous path planning problems. Specifically, we first employed DP to find the optimal solution for the center of each grid cell and then we created a novel method to compute the optimal solution for any continuous start position in the continuous workspace using only local information of neighbor cells. As a result, our method can achieve the optimal solution for any start position during robot continuous navigation compared to previous works using DP-based path planning methods.

Although DP-based data collection can provide optimal experience training data for DRL, it is likely that the learning process will still be too long or even get trapped in a local minimum and fail to converge if the data are fed to the DRL algorithm directly. To further improve the convergence performance and speed up the learning process, we propose the use of extreme learning machines (ELM) prior to the DRL in order to start DRL at a near-optimal starting point. This staged learning method allows DRL to fully focus on the most challenging part of the path planning, such as the areas around the obstacles or the ridges along which multiple optimal paths diverge. This initialization can be very fast due to the time efficiency feature of ELM.

By using DRL in this study, we aimed to learn the optimal action policy, i.e., closed-loop feedback for the real-time navigation of mobile robots in a 2D environment. Our optimal closed-loop action policy, which applies to any start position and covers all trajectories in the free workspace, is better than the optimal open-loop action sequence that is restricted to a single start position and therefore one trajectory. The optimal action policy allows for a real-time, fast, optimal response as the robot moves around in a complex environment. The closed-loop action policy eliminates the shortcomings of open-loop action sequences. It is also robust to disturbances or noise and reduces the deviations caused by disturbances since the effects of the disturbances are automatically compensated for. For example, in the case of the robot's state deviating from the optimal path caused by disturbance, the closed-loop action policy can mitigate the deviation from the original trajectory without accumulative errors. The optimal action policy can guarantee that the robot can still move along a new optimal path starting from the new disturbed state or position, which is close to the original optimal trajectory while the open-loop action sequence can cause the robot to move far away from the original optimal trajectory due to accumulative errors. Therefore, the closed-loop optimal policy is crucial for the real-time optimal navigation for mobile robots.

The contributions of this work are multifold: First, we mapped the dynamic programming method to typical optimal path planning problems for mobile robots and created an efficient dynamic programming-based method to find an exact, analytical, optimal solution for the continuous path planning problem. We then used high-quality training data gathered from the dynamic programing method for DRL, which greatly improves data quality and learning efficiency. We established a new two-stage learning method where we employed ELM to learn from initial optimal experience data in order to initialize the actor and critic neural networks to a near-optimal solution prior to DRL. Finally, we illustrated our proposed method using typical path planning tasks.

We organized the paper as follows. In Section 2, we provide an overview of the most relevant and recent papers in the fields of path planning, dynamic programming, optimal path planning, and DRL for path planning. In Section 3, we detail the new DRL approach implemented for this study. Experimental results are presented in Section 4. We conclude the paper and offer a few pointers for future research in Section 5.

## 2. Related Work

Path planning is a research field studying the moving strategies of robots or vehicles. In path planning, the goal is to safely move an agent from a start position to its corresponding final destination while evading any obstacles or other agents [12–15]. With numerous applications, such as improving robot movement efficiency and safety, increasingly efficient and powerful algorithms have been developed. To this end, research has been focused on optimal path planning, which can be defined with one or more of the following constraints: shortest time [20], shortest distance [21], and optimal energy consumption [22].

Conventional path planning methods, such as RRT, A*, and Bug approaches [13,14,23–26], aim to find one path from only one start position to the goal position, i.e., an open-loop sequence of points along the trajectory. When the start position changes or the mobile robot deviates from the optimal path caused by disturbances, path planning is reinitialized from the new start position. In contrast, our proposed method produces a closed-loop optimal action and control policy. With this policy, once the continuous optimal policy is learned via DRL, the robot can always move along the optimal path starting from any new position even when disturbed in the free workspace, based on the optimal policy and without replanning. Therefore, the purpose of our proposed approach is different from RRT, A*, and Bug approaches; our approach is better.

Deep learning (DL) is a new learning paradigm that performs nonlinear transformation using a multi-layer network structure [11,27–33] and has been gaining popularity in the last decade. In [28], Yann LeCun et al. presented a landmark work that employed the back-propagation method to acquire kernel coefficients directly from imaging representations of human written numbers, resulting in automated learning. We can categorize DL into three main groups: supervised learning [29], unsupervised learning [30], and reinforcement learning [31]. Unsupervised learning aims to find the hidden structures in unlabeled data but is ineffective in path planning problems due to the nature of this type of learning. On the other hand, supervised learning is much better with path planning problems due to its ability to easily converge and its lack of need to specify how the task should be performed. However, it is not without any flaws; in some applications, it can be hard to collect enough labelled data. In addition, the performance of this method is also limited: the robot cannot outperform the "supervisor", i.e., the training data, in supervised learning. In comparison, the last main method, DRL, does not need labelled data for training and can adequately generalize to new scenarios. However, DRL suffers the drawback of low sample efficiency [32].

In the last decade, DRL has been successfully applied to more and more applications. DRL performs better than human players in various fields by trying different strategies. In particular, DRL has been successfully used in mobile robotics or automated vehicle motion planning [1–11]. For example, in [10], the authors investigated the performance of a DRL-based deterministic policy gradient method for the dynamic environment. The algorithm was applied to multiple vehicle path planning. In [2], the authors studied a complex, big environment in which conventional algorithms often result in failures. They used a DRL-based algorithm to map vehicle's control actions to sensory inputs [2]. In [3]. The authors focused on a mixed environment and used DRL for vehicle motion planning. This algorithm was used for automated, multi-vehicle scenarios. In addition, by understanding and predicting the motions of moving obstacles using the equipped sensors, DRL can be programmed to avoid dynamic obstacles [4]. The elements of social interaction were introduced to DRL [5], where social rules were used to guide DRL. When we consider non-communicating multi-agent path planning problems, traditionally, the computation time can be prohibitively long due to unobservable agents' goals. However, DRL can be effectively used to reduce online computation time [6] and enable real-time navigation. While it is more challenging for DRL to avoid collision when the dynamic obstacles or agents do not follow any behavior rules, we can mitigate this issue by adding an LSTM segment for flexible observation size. This allows the DRL algorithm to achieve a better performance as the number of mobile obstacles increases [7]. A proposed method with

low computation time and low energy consumption while achieving optimal or close to optimal path formulates a clustered IoT network as a combinatorial optimization problem. The authors in [8] used a seq2seq network and DRL to train this network using information of the clusters and the UAV's start or end as the input. In [9], the authors introduced a new combination of elite-duplication genetic-assisted path planning with DRL. Using this method, they can optimally generate sparse waypoints in a constrained space. In practical path planning applications, such as for warehouse robots, there are multiple agents. To address this problem, in [10], the authors formulate the problem as a decentralized, partially observable Markov decision process and use a DRL approach to solve the problem by feeding global and local map representations into convolutional layers.

In the literature, many works have been presented in the research area of dynamic programming and DP-based optimal path planning. However, we found that most of these works cannot offer an exact continuous solution to the global optimal problem. They tend to use a distance transformation algorithm. This approach aims to find the paths from the goal position back to the start position. In order to use this method, we generate a distance wave front, which is propagated to cover all free space beginning from the destination [16–19]. The authors in [13] presented a constrained traveling distance transformation algorithm, which can search for the shortest distance between any two points with the presence of static obstacles. This method calculated the optimal traveling distance by discretizing the workspace into image pixels and approximating the traveling distance to the closest pixel of reference for every grid point. The authors in [17] extended the idea of the distance transform method for 2D path planning. They defined the propagated cost as a weighted sum of the traveling distance to the destination and the total cost of obstacles moving closer. The authors in [18,19] presented a real-time obstacle avoidance path planning for robots, which is applicable to scenarios with dynamic targets and obstacles. In this particular path planning scenario, the authors aimed to minimize both the cumulative local penalty functions along the path and the sum of the current known distance to a target. In all of the above works, the environment was discretized; researchers could not find an exact, analytical, continuous optimal solution for the path planning problem.

In [34], the authors proposed an actor-critic deep reinforcement learning method with experience replay. The sampling method described in this paper is very efficient. In [35], the authors proposed a Q algorithm-based ELM (Q-ELM) to tackle a slow convergence problem. In this ELM, the input was the mobile robot's perception of the external environment information and the output was the corresponding reward and punishment for each action decision, which was the Q value.

The essence of conventional extreme learning machines (ELM) [36–44] is to use a single hidden layer feedforward neural network for training and learning. Later, ELM has been extended to use neural networks with multi-layer hidden nodes for different applications [36]. Since hidden nodes do not need to be iteratively tuned, ELM learns much faster and yields more promising performance compared to multiple layer perceptron (MLP)-based algorithms [37]. ELM can also provide a unified learning platform with a widespread type of feature mappings and can be applied in regression and multiclass classification applications [38]. However, this method is not flawless. Compared with MLP, in order to achieve comparable accuracy, ELM often needs much more hidden nodes [39]. In [40], the authors directly used ELM in a path planning method. They use a multi-layer ELM to calculate the cost function of the A* algorithm and determine the accurate search direction by evaluating the impact of obstacles. The authors in [41] designed an adaptive fuzzy neural network planning method based on ELM. The ELM is used to solve classification and regression problems and is applied to quickly and accurately reduce the computational complexity of the traditional adaptive neural network. In this work, we used ELM to initialize DRL actor and critic neural networks, which were able to quickly produce the solution to the vicinity of the global optimum due to its short computation time and high-quality optimal initial training data.

## 3. Methodology

In this study, we evaluated our approach in a multi-obstacle environment, a typical layout for path planning applications, such as warehouse fulfillment. In this scenario, we assume that the robot can accurately measure the distance between itself and its surrounding environment with common sensors, such as Lidars and cameras.

Our main aim was to enhance the performance of learning algorithms and training data quality associated with DRL. The two basic steps in DRL are the data collection and the training process. At the data collection step, the traditional DRL usually collects random samples in the form of $(s_t, a_t, r_t, s_{t+1})$. These random samples are generally of low quality and result in a much longer training process in the learning stage. To solve this problem, we propose using DP that can find the global optimal trajectories from any start position to generate high-quality training data. In the following Section 3.1, we show how the path planning problem was formulated into a DP problem and solved for optimal paths, which in turn provides best quality data for DRL training. Conventionally, during the training step, the robot uses DRL algorithms directly to learn the optimal paths. In this Section 3.2 of this paper, we propose and discuss a detailed, novel two-stage deep reinforcement learning algorithm for fast learning.

Specifically, DP-based training data contain the global optimal information. The derived optimal experience data include not only local information but also key global useful information, which can effectively guide the reinforcement learning process. Therefore, the learning process can efficiently learn from these best experience data and converge fast. For example, global optimal moving directions (i.e., actions) are the highest quality information from the environment and are much more useful for learning than low-quality randomly collected experience data.

The major challenge of complex continuous action policy representation using the neural network is the tendency to fall in a local minimum. ELM can effectively deal with this problem in our application scenarios of this study. ELM is one type of supervised learning and is much more efficient at learning than general reinforcement learning algorithms since the target information is generally not available in conventional deep reinforcement learning approaches. The reasons are as follows:

- DP-based optimal training data provide global optimal moving directions or actions and can be used as the optimal learning target.
- Given the input-to-hidden parameters of the actor and critic neural networks in DRL, ELM formulates the learning problem as a quadratic optimization problem, which has the closed-form solution, resulting in rapid non-iterative learning. More importantly, ELM can achieve the global optimal solution, which effectively reduce or overcome the local minimum problem that is inherent in DNN learning. Therefore, ELM provides an excellent starting point close to the global optimal solution for DRL algorithms.

### 3.1. Global Optimal Solution for Discrete Grid Cell Centers Using the DP Method

In this section, we present a two-step DP-based path planning method. This method uses a distance propagation method to find the shortest distance from any position in the workplace to its corresponding destination. Specifically, we separated the task into two steps: a path planning step and a robot navigation step. During the first step, we divided the workspace into grid cells and use a DP-inspired algorithm to find the shortest distance from each cell center to the goal position. Next, during the navigation step, we generated continuous optimal trajectories from any start position in the workspace by using only the local information of neighboring cells.

### 3.1.1. DP-Inspired Algorithm for Global Optimal Path Planning for Discrete Grid Cells

In this study, we began the planning step by first partitioning the environment into grid cells and setting the goal with zero distance. We then calculated the distances from all the neighbor cells of the goal, eight in total, to the final goal position. Specifically, the shortest distance from each cell center to the destination was calculated, as shown in

Figure 1. Next, this distance generation method propagated through all the remaining cells, finally obtaining the shortest distance for each cell. A map was built for the whole workspace.

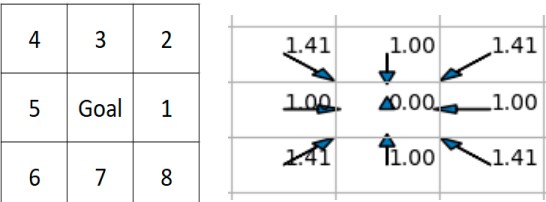

**Figure 1.** Goal and its eight neighbors (**left**) and shortest distance of the neighbor cells to goal (**right**).

We store the information for each cell, such as the center of the cell, the shortest length from the center of the cell to the goal, the optimal moving direction for the cell, and flags, to show the information, such as whether the cell is (1) in an obstacle, (2) already visited, or (3) in the priority queue (PQ).

The state space in our method is continuous in the navigation step. Any point in the free workspace is associated with the shortest distance from the final goal calculated by using only local information. In contrast, in this planning step, we used a discretized map of cells for planning. Based on the idea of dynamic programming, we illustrate in Figure 2 the cells that are organized by layers in our proposed method. Each cell is associated with the shortest length from the center of the cell to the goal. Each cell also contains key information, such as the optimal moving direction, which is used to guide the robot to the destination along the optimal path. In our method, only local information provided by its valid, available neighbors is required to calculate the shortest distance of a cell. We define valid neighbor cells as those that are not in any obstacle and define the available cells as the ones with the shortest lengths, which are already calculated from the previous steps.

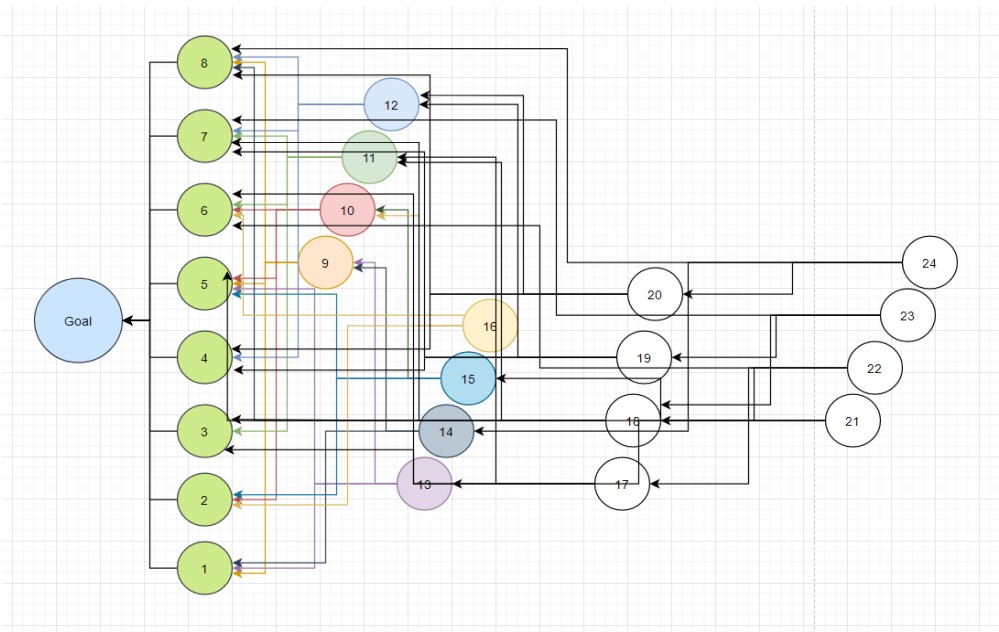

**Figure 2.** The illustration of the DP-like, layered-structure representation of the cells.

The DP-based formula for calculating the shortest traveling distance is as follows:

$$V_k(X_k) = \min_{X_{nbr} \in D_{nbr}(X_k)} \{d(X_k, X_{wpm}) + V_{k+1}(X_{wpm})\} \tag{1}$$

$$X_{wpm} = X_{nbr} + V_d(X_{nbr}) \tag{2}$$

where $d(X_k, X_{wpm})$ stands for the Euclidean distance from $X_k$ to $X_{wpm}$ and $V_{k+1}(X_{wpm})$ is the shortest distance to the goal. $X_{nbr}$ is the center of $X_k$'s neighbor cell, $X_{wpm}$ is the corresponding waypoint of $X_{nbr}$, and $V_d(X_{nbr})$ is the optimal moving direction at $X_{nbr}$. As for a valid neighbor cell $X_{nbr}$, notice that there is no known obstacle between $X_k$ and $X_{wp}(X_{nbr})$, i.e., the line segment of $X_k$ and $X_{wp}$, does not pass any obstacle.

The global optimal direction $V_d$ from the cell center $X_k$ can be calculated as follows:

$$V_d = X_{wpm*} - X_k \tag{3}$$

where $X_{wpm*}$ is the optimal waypoint that produces the shortest traveling distance $V_k(X_k)$, which is obtained using Equation (1).

Figure 3 shows a general case of how the shortest distance is determined for each cell. For example, there are three valid available neighbor cells for cell 326, which are cell 315, cell 322, and cell 312. We can compute the shortest distance and the optimal moving direction for cell 326 based on only the local information of these three surrounding cells.

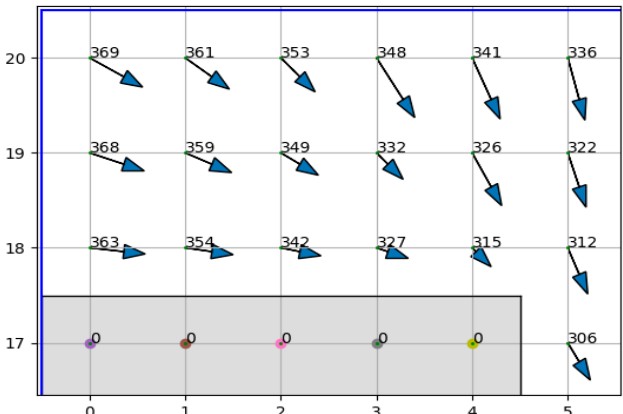

**Figure 3.** Calculation of the shortest distance for the cell with three valid neighbors available.

Note that not all surrounding cells have a valid shortest distance due to the presence of the obstacles. The center of a cell may be inside an obstacle and may have no valid value. As long as we can find one or two neighbors of the cell close to the obstacle with valid shortest distance values, we can still calculate the shortest distance for that cell. Lastly, it is also important to check whether the line connecting the cell and the waypoint passes through any obstacle.

In Figure 4, we show how to employ only local information to obtain the global shortest distance of a cell to the goal. In this example, the shortest distance of cell 37 is calculated using only the information of three neighbors: cell 13, cell 21, and cell 29. We also calculate the corresponding optimal waypoint $P_{wp}$ using the centers and optimal moving directions of the neighboring cells using Equations (1) and (2). The shortest distance $V(P_{c37})$ is then calculated:

$$V(P_{c37}) = d(P_{c37}, P_{wp}) + V_{k+1}(P_{wp}) \tag{4a}$$

where $V_{k+1}(P_{wp})$ stands for the shortest distance from $P_{wp}$ to the goal. $d(P_{37}, P_{wp})$ represents the distance between the center $P_{c37}$ of cell 37 and the waypoint $P_{wp}$. In contrast with other methods [3], the optimal traveling distance from cell 37 does not need to pass through the center of neighbors 13 or 21; instead, it can find the optimal, accurate moving direction.

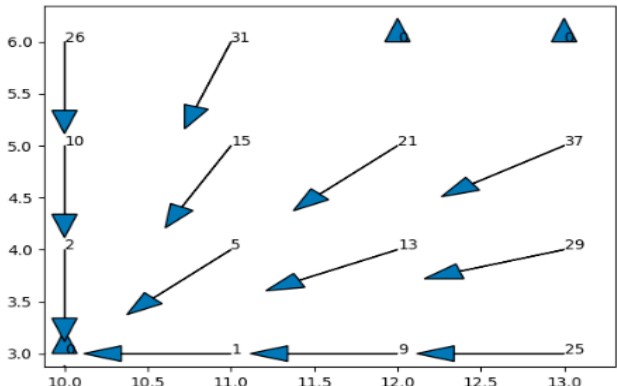

**Figure 4.** Computing global shortest distance of cell 37 from local neighbors: cells 13, 21, and 29.

In the implementation of our algorithm, we used a priority queue (PQ) to choose the next cell to consider and find the shortest distance from the cell to the goal. This process mimics the wave front propagation from the goal position. The cell priority $P_{dist}(C)$ was computed as follows:

$$P_{dist}(C) = \min_{C_j \in D_{nbr}(C)} \{d(C, C_j) + V_{k+1}(C_j)\} \qquad (4b)$$

where $d(C, C_j)$ represents the Euclidean distance from the center of cell $C$ to the center of cell, $C_j$, $C_j$ is the valid available neighbor, $D_{nbr}(C)$ is the set of all the valid available neighbor cells of current cell $C$, and $V_{k+1}(C_j)$ is the shortest distance of $C_j$ to the goal position. In Figure 5, we show the progress of distance propagation. In Figure 5a, we show the result of the propagation after 44 cells. In Figure 5b, we show the result of the propagation after 371 cells.

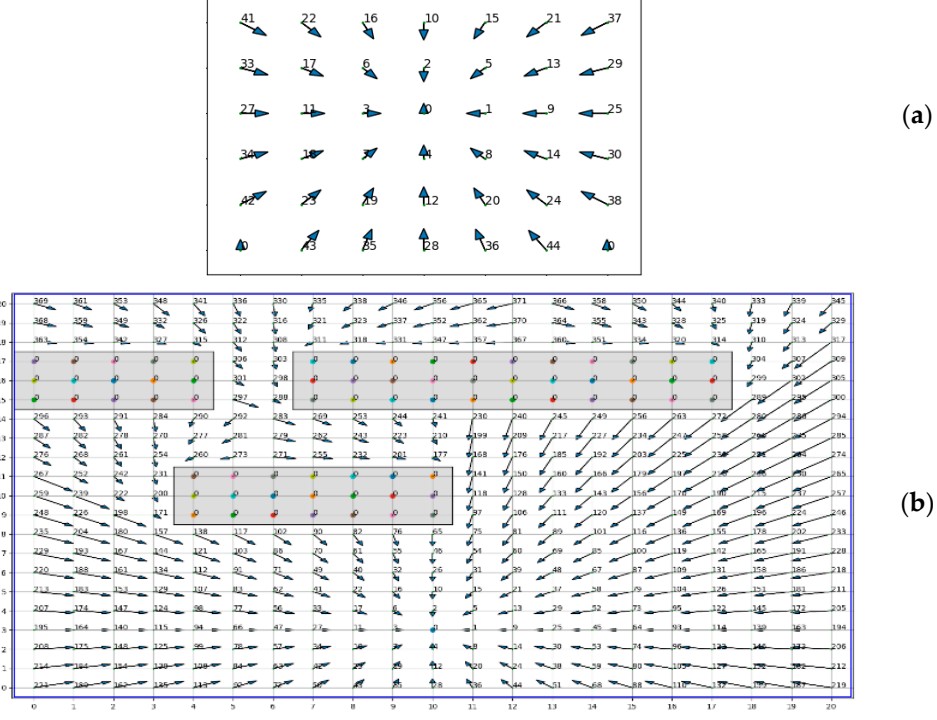

**Figure 5.** Progress of shortest distance propagation: (**a**) visited 44 cells, (**b**) visited 371 cells.

In this study, we developed a DP-based shortest traveling length of path planning algorithm, as shown in Algorithm 1.

---

**Algorithm 1.** DP-Inspired Shortest Distance Path Planning Algorithm

---

Step 1. Initialize the environment:
Step 1.1. Partition the whole map into $N \times M$ grid cells;
Step 1.2. Set cell properties for obstacles such as obstacle flag;
Step 1.3. Set the goal cell with zero distance.
Step 2. Process eight neighbors of the goal cell. For each neighbor, do the following operations:
Step 2.1. Calculate the shortest distance from the goal to the center of the cell, set cell properties such as the shortest distance, optimal moving direction, visit flags;
Step 2.2. Check each of eight neighbors of the current cell, add it to priority queue (PQ) if (not in PQ) AND (not visited yet) AND (not obstacle).
Step 3. Cell propagation for the whole free workspace:
Step 3.1. Take the top cell in PQ;
Step 3.2. Compute for the cell the shortest distance and optimal moving direction using only the local information of available, valid neighbor cells using Equations (1) and (3);
Step 3.3. Set cell properties such as the shortest distance, optimal direction, visit flags;
Step 3.4. Check each of eight neighbors of the cell, add it to priority queue (PQ) if (not in PQ) AND (not visited yet) AND (not obstacle);
Step 3.5. Back to step 3.1 until PQ is empty.
Step 4. Check and mark the cells close to ridge boundary based on local information of neighbor cells.

---

### 3.1.2. Navigation Step: Exact Shortest Distance Calculation for Any Point in the Map

This navigation step is a real-time process in which the robot travels along the optimal traveling path from any point in the continuous map. The trajectory does not need to go through grid cell centers. We calculated the optimal direction for any point using only local information of the neighbor cells. We discuss two cases in detail here.

In the first case, as shown in Figure 6a, the optimal moving directions of the valid neighbor cells are all pointing towards the same waypoint. We calculated the waypoint $P_{wp}$ based on the information of any valid neighbor cell, the center position, and optimal moving direction of the cell. Then, we calculated the optimal traveling direction $V_p$ and the shortest traveling distance $d(P)$ from the current point $P(x,y)$ as follows.

$$V_p = P_{wp} - P \tag{5}$$

$$d(P) = d(P, P_{wp}) + V_{k+1}(P_{wp}) \tag{6}$$

where $V_{k+1}(P_{wp})$ is the shortest distance from $P_{wp}$ to the goal and $d(P, P_{wp})$ is the distance between points $P$ and $P_{wp}$.

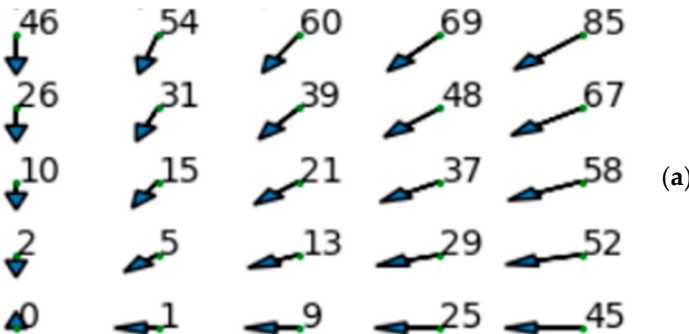

**Figure 6.** *Cont.*

**Figure 6.** Calculation of Exact Shortest Distance from Any Point in the Map. (**a**) Case I, (**b**) Case II.

In Case II, we considered the robot position *P*, which is close to the ridge boundary as shown in Figure 6b, where there are several waypoints in the neighborhood. Some potential, optimal directions guide the waypoint on the left while others guide the waypoint on the right. Similar to Case I, we first calculated the waypoint set $D_{wp}(P)$ from all valid neighbors. Next, we minimized the distance from the point to the goal in order to obtain the best waypoint as follows.

$$d(P) = \min_{P_{wp} \in D_{wp}(P)} \left\{ d(P, P_{wp}) + V_{k+1}(P_{wp}) \right\} \tag{7}$$

Then, we calculated the shortest distance $d(P)$ from the point *P* to the goal using the best waypoint $P_{wpm}$ and we also computed the corresponding optimal moving direction.

In this navigation step, for any arbitrary position in the environment, there are up to 9 valid cells in the neighborhood, including the cell containing the current point plus 8 neighbor cells. However, if the cell is not in the free space, some neighbor cell properties may not be available.

### 3.2. Two-Stage DRL

Although DRL has the potential to achieve superhuman performance in theory, in practice, it is very challenging to efficiently learn the parameters of the optimal continuous action policy, i.e., the continuous actor network for specific applications using DRL. Without an adequate initialization, DRL often converges too slowly or it may not even converge at all. In this paper, we propose a novel two-stage DRL algorithm based on the modified DDPG algorithm to efficiently learn the optimal policy for the mobile robot navigation. At the first stage, we employed the DP-based technique to generate many global optimal trajectories or experiences in the workspace. Based on these optimal trajectories, ELM was then used to calculate the favorable initial parameter values of the actor and critic networks (DNN) due to its non-iterative high computation speed and high-quality initial optimal training data. In the second stage, once the favorable initial values of the parameters are in the vicinity of the global optimum, DRL was used to fine-tune and ensure the high accuracy. More specifically, we employed both local experience data and global optimal experiences to guide the learning process to make the learning process converge to the optimal solution while avoiding the collision with obstacles using more local experiences in complicated regions, such as the regions close to obstacles and ridges. In the free workspace far away from obstacles or ridge regions, the optimal policy is relatively smoother and requires the smaller amount of training samples. By using this two-stage DRL method, we can significantly accelerate the learning process, reduce the probability of the robot getting trapped into a local minimum, and achieve the high-quality navigation policy.

### 3.2.1. Initial and Online Data Collection

In this study, the training data were partially collected prior to DRL training and partially during the training process. Before training, (I) the DP-based navigation approach was employed to generate multiple optimal trajectories by specifying random start positions and (II) for each grid cell, we produced experience data with optimal moving directions, obtained as described in Section 3.1. Experience data for the critic $Q(s,a)$ were then generated as follows. In the initial data collection, given a position or state *s*, there are two methods to produce the experience for $Q(s,a)$: (a) using the corresponding optimal moving direction *a* to advance one step to obtain the next state $s_{t+1}$ and compute the optimal $Q(s,a) = -d(s, s_{t+1}) + V^*(s_{t+1})$ or (b) randomly generate a moving direction and advance one step to obtain

the next state $s_{t+1}$ and compute $Q(s,a) = -d(s,s_{t+1}) + V^*(s_{t+1})$, where $V^*(s_{t+1})$ is the negative shortest distance from state $s_{t+1}$ to the goal. These trajectories were sampled to obtain initial training data. During training, further samples were collected and used in the later training process as the robot moves around in the environment and moves towards to the goal.

The data collection around the obstacles is particularly challenging as the robot (1) cannot collide with the obstacles and (2) needs to follow the optimal path, which often requires the robot to move closely around the obstacles. In order to avoid collision with obstacles, more samples were collected from the regions close to obstacles. Some samples are generated by advancing one step along the optimal moving directions while other samples are generated by advancing one step along random moving directions without collision with obstacles.

### 3.2.2. Using ELM for Near-Optimal Initialization

ELM learning was employed to rapidly initialize the actor and critic networks in deep reinforcement learning due to its short learning computation time and high-quality solution. Due to the fast computation, we can run ELM multiple times and select the best solution among multiple runs as the initial values of the DRL network parameters. ELM is able to achieve the global optimal solution for the quadratic programming problem for the given weights of input-to-hidden layers if the networks is with one hidden layer. Therefore, ELM initializes the parameters of the networks to a near-optimal solution when the training data are produced along the optimal moving directions generated in Section 3.1. By using ELM and global optimal initial training data, we also significantly reduced the probability of getting trapped in a local minimum.

Given a set of $N$ distinct training samples $\{(x_i, t_i) \mid x_i \in R^d, t_i \in R^m, i = 1, 2, \ldots, N\}$ where $x_i$ is the training input data vector, $t_i$ represents the target of each sample, and $L$ denotes the number of hidden nodes, one single hidden layer neural network with $L$ hidden neurons can be written as [39]:

$$y_i = \sum_{j=1}^{L} \beta_j g(w_j x_i + b_j) = \sum_{j=1}^{L} \beta_j h_j(x_i) = t_i + \varepsilon_i, \ i = 1, 2, \ldots, N \tag{8}$$

where $g$ is the activation function, $w_j$ and $b_j$ are random weights and biases, and $\varepsilon$ is noise or error.

The matrix form of ELM is presented here:

$$\min_{\beta} \|H\beta - T\|^2 \tag{9}$$

$$\beta = \begin{bmatrix} \beta_1^T \\ \ldots \\ \beta_L^T \end{bmatrix} = \begin{bmatrix} \beta_{11} & \cdots & \beta_{1m} \\ \vdots & \ddots & \vdots \\ \beta_{L1} & \cdots & \beta_{Lm} \end{bmatrix}$$

$$T = \begin{bmatrix} t_1^T \\ \ldots \\ t_N^T \end{bmatrix} = \begin{bmatrix} t_{11} & \cdots & t_{1m} \\ \vdots & \ddots & \vdots \\ t_{N1} & \cdots & t_{Nm} \end{bmatrix}$$

$$H = \begin{bmatrix} h(x_1) \\ \ldots \\ h(x_N) \end{bmatrix} = \begin{bmatrix} h_1(x_1) & \cdots & h_L(x_1) \\ \vdots & \ddots & \vdots \\ h_1(x_N) & \cdots & h_L(x_N) \end{bmatrix} \tag{10}$$

where $H$ is the hidden layer output matrix and $T$ is the training data target matrix. The above quadratic optimization problem can be solved in a closed form: $\beta = H^+ T$, $H^+$ is the Moore–Penrose generalized inverse of matrix $H$. Therefore, ELM is a non-iterative learning algorithm and is extremely fast.

During the ELM training phase, only the output weights $\beta$ are adjusted according to the algorithm. The ELM training algorithm can be summarized in Algorithm 2.

| **Algorithm 2.** Fast ELM Learning Algorithm |
| --- |
| Step 1. Randomly assign the hidden node parameters, i.e., the input weights $w_j$ and biases $b_j$ for additional hidden nodes $j = 1, 2, \ldots, L$ in Equation (8). <br> Step 2. Calculate the hidden layer output matrix $H$ using Equation (10). <br> Step 3. Compute the output weight vector $\beta$ as follows: <br> $$\beta = H^+ T \qquad (11)$$ |
| where $H^+$ is the Moore–Penrose generalized inverse of matrix $H$. |

Note that this algorithm is used in Algorithm 3 to initialize the actor and critic neural networks, where the parameters $\theta^Q$ or $\theta^\mu$ each includes $\beta$, $w_1, w_2, \ldots, w_L$, $b_1, b_2, \ldots, b_L$.

### 3.2.3. Actor and Critic Neural Networks in DRL

The actor network and critic network consist of three layers each, using the sigmoid activation function in the hidden layer. In the actor network, the input is the two-dimensional state vector, i.e., the position $s = (x,y)$ in the workspace, and the output is a two-dimensional action vector, i.e., the moving direction $a = (vx,vy)$. Meanwhile, for the critic network, the input is a four-dimensional vector $(s,a) = (x,y,vx,vy)$ and the output is the negative distance $Q(s,a)$ from the current state $s$ to the goal, taking the current action $a$. Both the actor and critic networks can be efficiently initialized using the rapid ELM algorithm.

### 3.2.4. Modified DDPG (MDDPG) for Fine-Tuning DRL Actor and Critic Networks

In this section, we modify the deep deterministic policy gradient method (DDPG) for planning the optimal path for the robot. The robot is assumed to interact with the environment $E$ in discrete timesteps. At timestep $t$, the robot accomplishes three things: it takes an action $a_t$, moves one step from the state $s_t$, and receives a reward $r_t$. Both the action and state spaces are continuous in this section.

The action-value function depicts the expected return in state $s_t$ after taking an action $a_t$. We detail the Bellman equation as follows:

$$Q^\mu(s_t, a_t) = E_{r_t, s_{t+1} \sim E}[r(s_t, a_t) + \gamma Q^\mu(s_{t+1}, \mu(s_{s+1}))] \qquad (12)$$

The loss function is defined as:

$$L\left(\theta^Q\right) = E_{s_t \sim \rho^\beta, \; a_t \sim \beta, r^t \sim E}\left[\left(Q\left(\theta^Q\right) - y_t\right)^2\right] \qquad (13)$$

where the reward signal $r(s_t, a_t)$ is the negative travel distance for each time step and $y_t = r(s_t, a_t) + \gamma Q(s_{t+1}, \mu(s_{s+1}) | \theta^Q)$.

The critic $Q(s,a)$ is learned using the Bellman equation. The actor is calculated by following the chain rule to the expected return from the derivative of $J$ with respect to the actor parameters:

$$\nabla_{\theta^\mu} J \approx E_{s_t \sim \rho^\beta,} \left[ \nabla_{\theta^\mu} Q(s,a | \theta^Q)\Big|_{s=s_t, \; a=\mu(s_t | \theta^\mu)} \right] = E_{s_t \sim \rho^\beta,} \left[ \nabla_a Q(s,a | \theta^Q)\Big|_{s=s_t, \; a=\mu(s_t)} \nabla_{\theta^\mu} \mu(s | \theta^\mu)\Big|_{s=s_t} \right] \qquad (14)$$

### 3.2.5. Two-Stage DRL Algorithm

Next, the DDPG algorithm (MDDPG) is modified to include DP-based data collection before and during the training time. ELM is employed to initialize the parameters of the actor and critic networks as well.



---

**Algorithm 3.** Two-Stage DRL Algorithm

---

Step 1. Prior to MDDPG training, DP-based data collection is performed to obtain the initial optimal actor data and optimal critic data.

Step 2. Randomly generate initial weights of $\theta^Q$ and $\theta^\mu$ of actor and critic networks, respectively. Note that the parameters $\theta^Q$ and $\theta^\mu$ include $\beta, w_1, w_2, \ldots, w_L, b, b_2, \ldots, b_L$ in Algorithm 2.

Step 3. Based on initial training data, ELM is employed to rapidly compute hidden-to-output weights $\beta^Q$ and $\beta^\mu$ for the actor network and critic network, respectively. We then replace the corresponding parts with $\beta^Q$ and $\beta^\mu$ in the above randomly generated initial weights $\theta^Q$ and $\theta^\mu$.

Step 4. Initialization for MDDPG training:

Step 4.1. Initialize critic network $Q(\theta^Q)$ and actor network $\mu(\theta^\mu))$ with weights $\theta^Q$ and $\theta^\mu$, respectively;

Step 4.2. Initialize target network $Q'$ and $\mu'$ with weights $\theta^{Q'} = \theta^Q, \theta^{\mu'} = \theta^\mu$;

Step 4.3. Initialize replay buffer $R$ with $N_0$ collection steps with DP-based optimal experiences;

Step 4.4. Receive observation of start state $s_1$.

Step 5. Using MDDPG to fine-tune actor and critic neural networks

Step 5.1. Collect more training data for action exploration and stored in $R$;

Step 5.2. Select action $a_t = \mu(\theta^\mu) + N_t$ according to the current policy and exploration noise $N_t$;

Step 5.3. Execute action $a_t$ and observe reward $r_t$ and new state $s_{t+1}$;

Step 5.4. Store transition $(s_t, a_t, r_t, s_{t+1})$ in $R$;

Step 5.5. Sample a random mini-batch of $N$ transitions $(s_t, a_t, r_t, s_{t+1})$ from $R$;

Step 5.6. Compute.

$$y_i = r_i + \gamma Q'\left(s_{i+1}, \mu'\left(\theta^{\mu'}\right)|\theta^{Q'}\right) \tag{15}$$

Step 5.7. Update critic by minimizing the loss:

$$L = \frac{1}{N}\sum_{i=1}^{N}\left(y_i - Q(s_i, a_i|\theta^Q)\right)^2 \tag{16}$$

Step 5.8. Update the actor policy using the sampled policy gradient:

$$\nabla_{\theta^\mu} J \approx \frac{1}{N}\sum_i \nabla_a Q\left(\theta^Q\right)|_{s=s_i, a=\mu(s_i)} \mu(s|\theta^\mu)|_{s_i} \tag{17}$$

Step 5.9. Update the target networks:

$$\theta^{Q'} = \tau\theta^Q + (1-\tau)\,\theta^{Q'} \tag{18}$$

$$\theta^{\mu'} = \tau\theta^\mu + (1-\tau)\,\theta^{\mu'} \tag{19}$$

Step 5.10. Repeat Step 5.1 until reaching the maximum number of training or stop criteria.

---

## 4. Experimental Results

### 4.1. Environment

To evaluate the effectiveness of our method, we used our method in typical motion planning scenarios with multiple obstacles. The vehicle would be able to travel from any beginning position in the workspace and reach the goal position without colliding with any obstacles. One typical scenario is shown in Figure 7. The rectangle obstacles in the figure depict the barriers. The beginning position of the vehicle and the goal position are marked out in the figure.

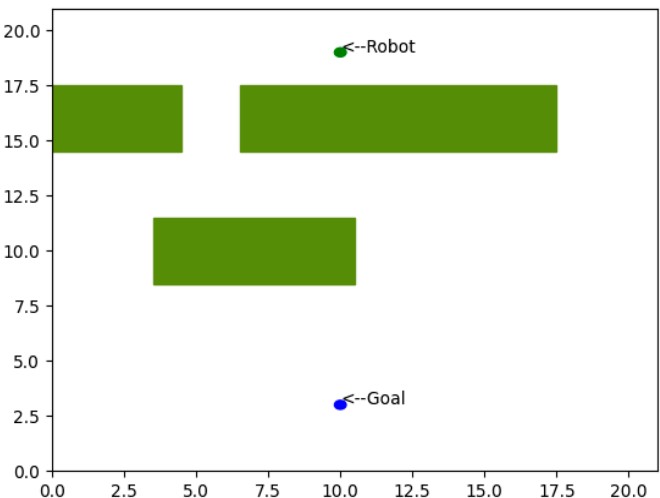

**Figure 7.** Robot working environment.

*4.2. Vector Fields and Shortest Travelling Distance of the Optimal Path Map*

To demonstrate the efficacy of the proposed method, we carried out the experiments for two typical scenarios with multiple obstacles. In Figure 8, we show the calculation order for the cell distance propagation. Every number before each arrow shows the order by which the cell is travelled. In Figure 9, we show the shortest travelling distance (i.e., the number before each arrow) from the goal position to each cell. In these two figures, every arrow represents the optimal moving direction for the center of the cell to the goal. The goal position is represented by a triangle with zero distance. Obstacles are represented by rectangular shapes. For any point in the free workspace, based only on the shortest distances and optimal moving directions of local neighbor cells as shown in Figure 9, we can efficiently calculate the corresponding optimal moving direction (i.e., action) and the shortest distance from the point to the goal position in order to navigate the robot to advance along the optimal path to the goal.

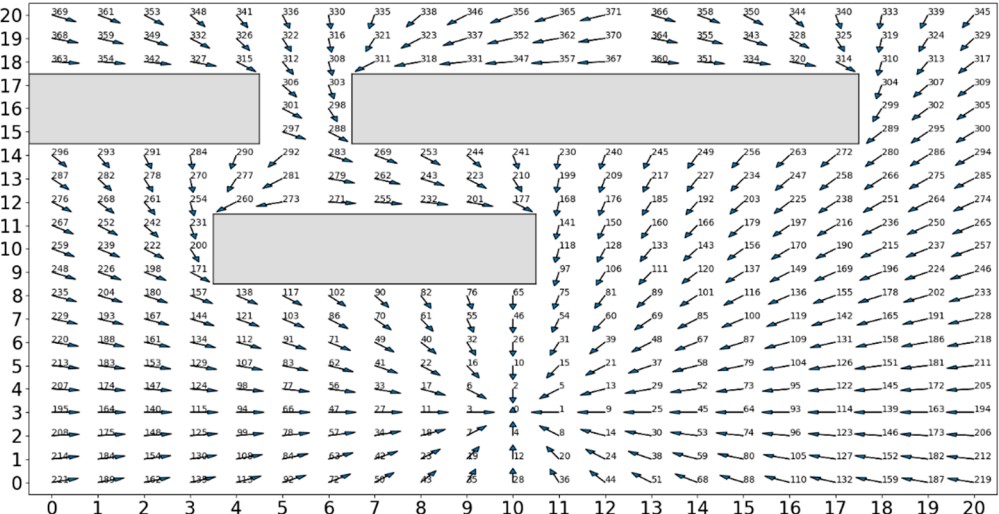

**Figure 8.** The computation order (each number before arrow) for the cell propagation.

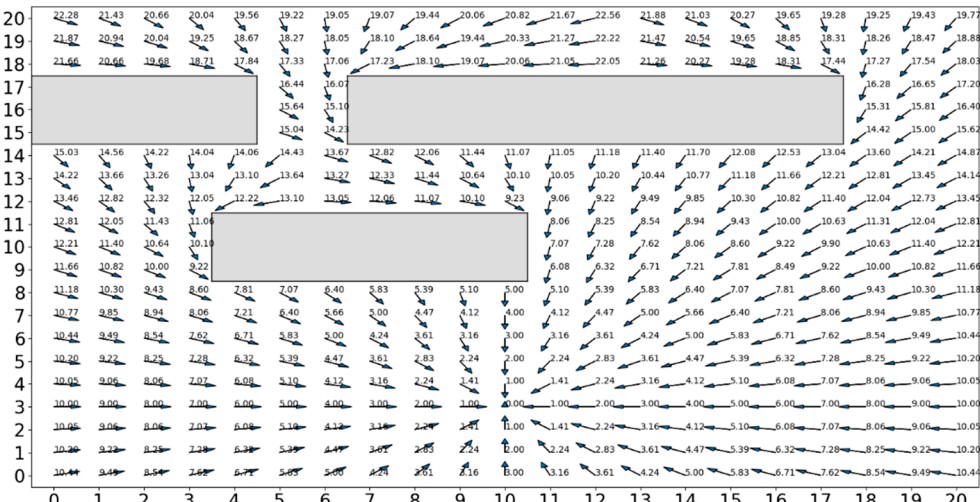

**Figure 9.** Shortest distance (the number before each arrow) and optimal moving vector fields of the map (x, y axes: indices of cells).

In Figure 10, we consider an environment with multiple obstacles with different shapes. We show the visitation order and the optimal moving vector fields of the entire environment.

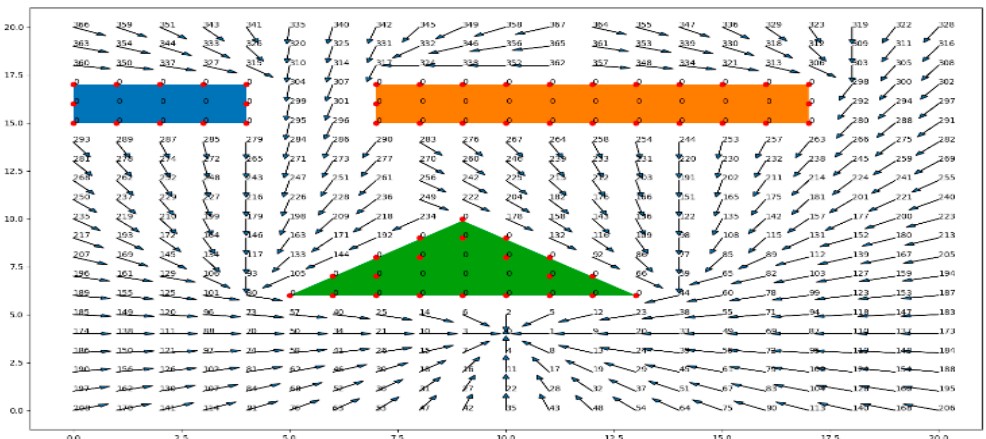

**Figure 10.** The visitation order and optimal moving vector fields of the map.

The Ridge Boundary

In Figure 11, we focus on the ridge boundary that separates two sets of moving directions. The starting points on the left of the ridge boundary take the paths to the far left and the starting points on the right of the ridge boundary take the paths to the far right. As a result, these starting points travel through different waypoints.

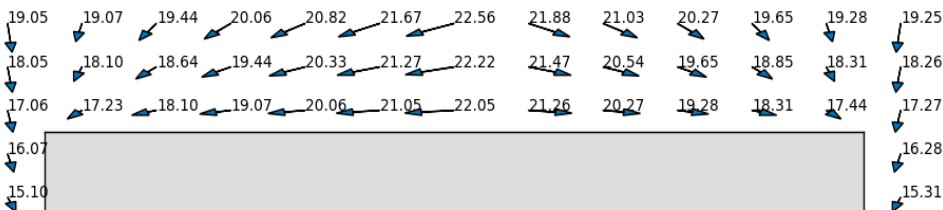

**Figure 11.** Ridge boundary in the middle of this figure to separate cells into left and right parts with different moving directions.

### 4.3. Sample Optimal Trajectories for Optimal Data Collection

By using the navigation algorithm described in Section 3.1.2, we randomly selected multiple starting points and computed their optimal trajectories, as shown in Figures 12 and 13. From any beginning point, the vehicle can move along the shortest traveling path all the time. These optimal trajectories can be sampled and fed into the two-stage DRL for training.

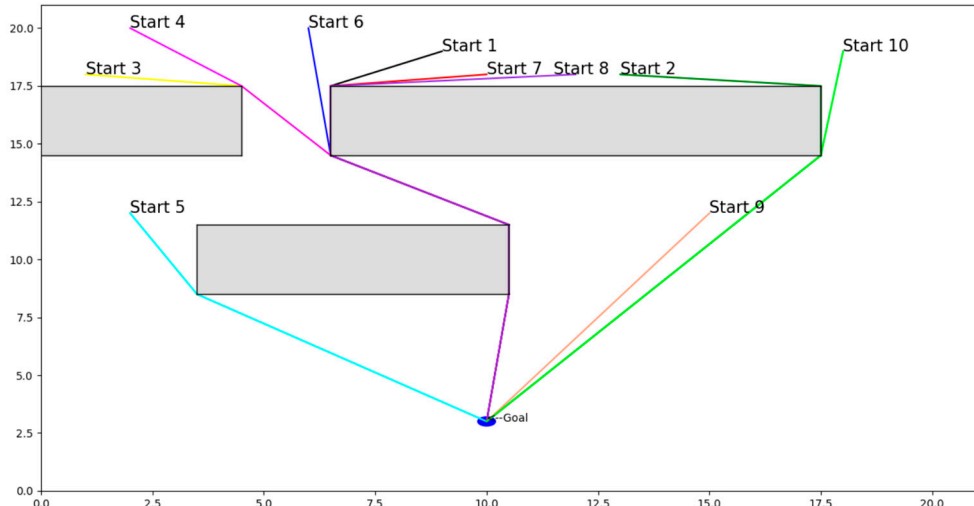

**Figure 12.** Typical sample optimal trajectories for data collection.

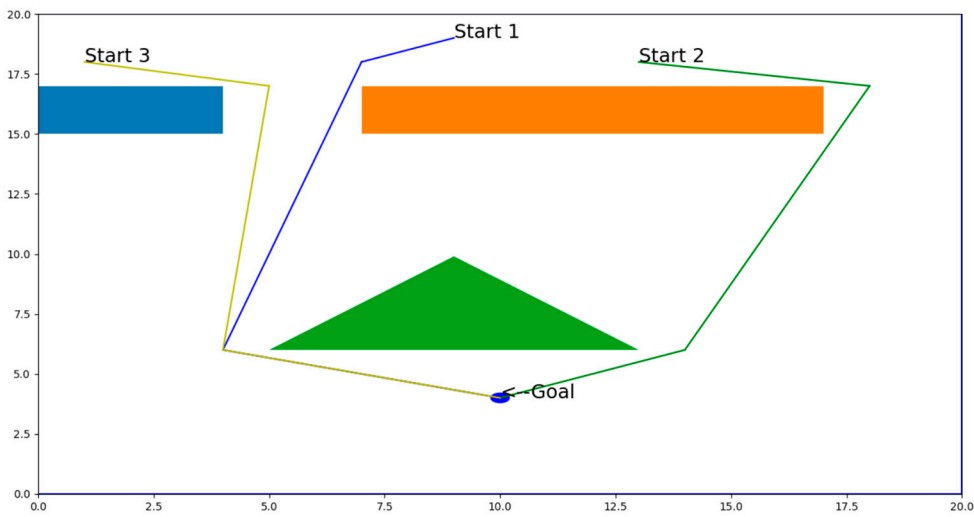

**Figure 13.** Sample trajectories for data collection in another scenario.

### 4.4. Sample Optimal Trajectories Generated by the Two-Stage DRL Algorithm

Once we collected the initial data of optimal trajectories using the DP-based navigation technique, we used our two-stage DRL to train the actor and critic neural networks. Instead of using random data for trial-and error-iterations, we used these high-quality data for initial training with ELM.

#### 4.4.1. After First Stage: ELM Learning

The sample trajectories after ELM learning are shown in Figure 14. ELM can provide a near-optimal starting point for the neural networks in a short time. However, some path segments near obstacles may not be accurate. This problem with ELM learning is shown in Figure 16. Thus, in the second stage, we can use the modified DDPG algorithm to further improve the training accuracy.

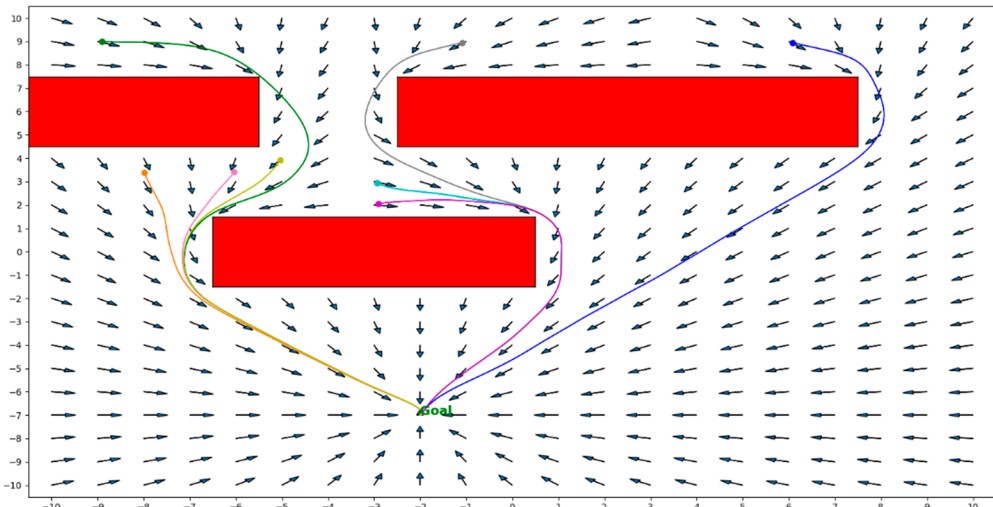

**Figure 14.** Sample trajectories after ELM initialization.

### 4.4.2. After Second Stage: DRL

The sample trajectories for one scenario after DLR learning are shown in Figure 15. In this stage, DRL focuses on the more challenging parts of the paths and regions. In Figure 15, we show that with DRL, we can improve the accuracy of the paths.

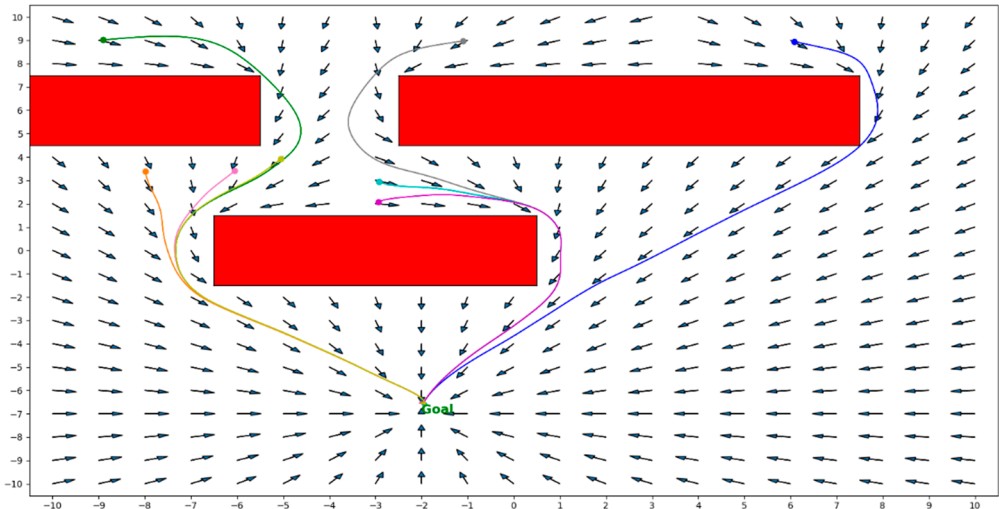

**Figure 15.** Sample trajectories after DRL fine-tuning.

In the regions close to obstacles, it is much more challenging to train the actor and critic networks. For example, regular experience data collection can lead to collision with the obstacle, which is shown by the green trajectory in Figure 16. After we added more experience data to the neighbor of the obstacle, the learned policy network was able to produce an optimal trajectory (blue line in Figure 16) while avoiding the collision with the obstacle.

In order to illustrate the advantages of our proposed DRL method over the original DDPG algorithm, we conducted the following experiment. In the same scenario as that depicted in Figure 15, we employed the DDPG algorithm to train the actor and critic neural networks with the same training experience samples. The only difference in this experiment using the original DDPG algorithm is that we employed random weights to initialize the actor and critic networks instead of using the weights from ELM learning results. We repeated the experiment for three times and they all failed to find the feasible paths. In Figure 17, we show a typical trajectory resulting from the original DDPG algorithm.

The robot starts from a position (−9.0, 9.0) and collides with the obstacles, even after 250,000 training iterations. The main reason for this failure is that, compared to supervised learning, deep reinforcement learning is much harder to converge, less efficient in learning, and easier to get stuck in local minima. These implementation issues associated with the conventional DRL algorithm result from many factors including weak guidance signals, obstacles, and long episodes in the complex environment. In contrast, our fast ELM learning with DP-based optimal data collection is able to initialize the corresponding weights to a near-optimal solution, which significantly improve the learning effectiveness and efficiency. In Figure 18, we show the trajectory of using our proposed method for the same scenario using only 70,000 training iterations. The robot can follow the optimal path, avoid obstacles, and reach the target successfully. This experiment demonstrated that our proposed method is much more efficient than the original DDPG algorithm.

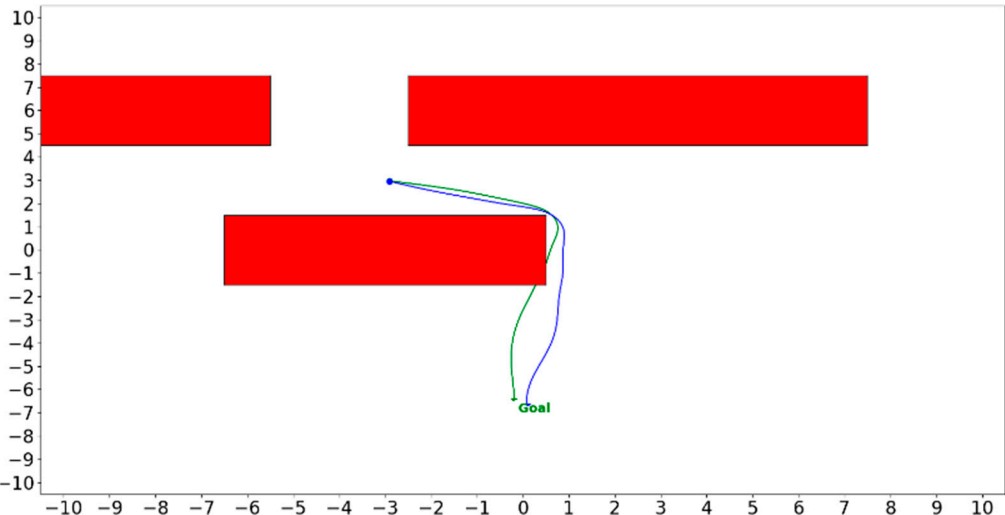

**Figure 16.** Collision with obstacles (**green**), collision avoidance by sampling more experiences close to obstacles (**blue**).

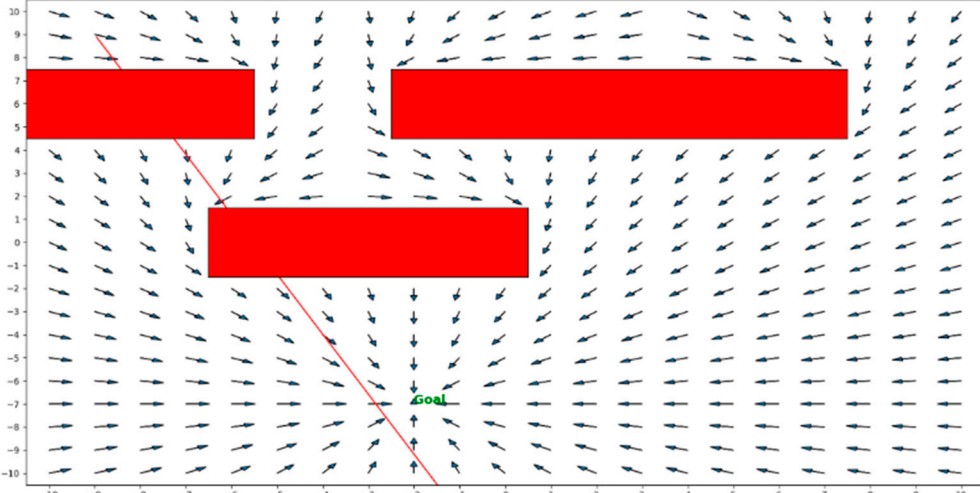

**Figure 17.** Collision with obstacles (red trajectory) with the conventional DDPG learning algorithm, even with the same training experience samples.

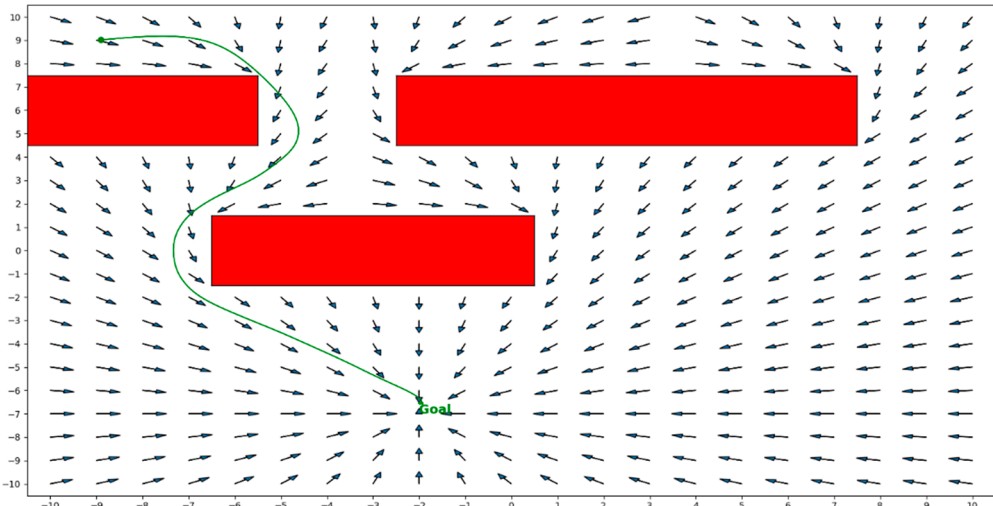

**Figure 18.** Trajectory produced by our proposed DRL method.

**5. Conclusions and Future Work**

In this paper, we presented a novel optimal path planning algorithm based on DRL with application to mobile robots. In order to improve training data quality and generate optimal training data for DRL, we mapped the DP method to typical optimal path planning problems and established an efficient DP-based method to find the exact, analytical, optimal solution. In order to accelerate the reinforcement learning process and improve the learning performance, we also created a two-stage DRL method, in which ELM was employed to initialize the weight parameters of the actor and critic networks. This algorithm is able to move the robot along an optimal path from any starting point in the continuous workspace to a specified goal location. For our next steps, we plan to conduct a comprehensive study to compare our proposed method with other existing techniques. We also plan on extending the capability of our algorithm to handle 3D environments and environments with obstacles of arbitrary shapes, moving obstacles, and multiple agents.

**Author Contributions:** Conceptualization, J.R. and X.H.; methodology, X.H., J.R. and R.N.H.; software, X.H. and R.N.H.; validation, J.R., X.H. and R.N.H.; formal analysis, X.H.; investigation, J.R. and X.H.; resources, X.H.; data curation, X.H.; writing—original draft preparation, J.R. and R.N.H.; writing—review and editing, X.H. and R.N.H.; visualization, R.N.H.; supervision, J.R.; project administration, J.R.; funding acquisition, J.R. All authors have read and agreed to the published version of the manuscript.

**Funding:** The authors would like to acknowledge the support of the Natural Sciences and Engineering Research Council of Canada (NSERC) [funding reference number 210471].

**Data Availability Statement:** Public datasets were not used in this study.

**Conflicts of Interest:** The authors declare no conflict of interest.

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
