# Peer review of "Efficient Deep Reinforcement Learning for Optimal Path Planning"

_electronics, doi:10.3390/electronics11213628_

Round 1
Reviewer 1 Report
In this paper, the authors propose a deep reinforcement learning method for optimal path planning for mobile robots using dynamic programming. The experimental results have shown that the method can guide the robots to reach the goal position with good convergence. This article has certain innovation. However, some issues still need to be improved:
(1) The abstract of this paper does not state the motivation for this study. Meanwhile, Instead of summarizing the idea of the algorithm, the authors highlight the contribution of this paper, and this way of exposition is questionable.
(2) In the introduction and related works sections, this paper focuses on deep reinforcement learning rather than on optimal path planning, not only the motivation of this study, but also the related works.
(3) The representations of the two algorithms do not agree.
(4) Some equations are unlabeled.
(5) In the environment section, the proposed method in this paper are validated. However, this experiment lacks quantitative analysis and comparison with other excellent algorithms.
Author Response
Thank you very much for all the comments. We highly appreciate your time and efforts to improve our paper. We have revised the paper based on your valuable feedbacks and we also address your comments here in this letter.
Reviewer 1:
In this paper, the authors propose a deep reinforcement learning method for optimal path planning for mobile robots using dynamic programming. The experimental results have shown that the method can guide the robots to reach the goal position with good convergence. This article has certain innovation. However, some issues still need to be improved:
(1) The abstract of this paper does not state the motivation for this study. Meanwhile, Instead of summarizing the idea of the algorithm, the authors highlight the contribution of this paper, and this way of exposition is questionable.
Answer: We have revised the abstract to address this comment. We copy the new abstract here for your convenience.
“In this paper, we propose a novel deep reinforcement learning (DRL) method for optimal path planning for mobile robots using dynamic programming (DP)-based data collection. The pro-posed method can overcome slow learning process and improve training data quality inherently in DRL algorithms. The main idea of our approach is as follows. First, we map the dynamic programming method to typical optimal path planning problems for mobile robots and present a new, efficient DP-based method to find the exact, analytical, optimal solution for the path planning problem. Then, we use high quality training data gathered using the DP method for DRL learning, which greatly improves training data quality and learning efficiency. Next we propose a two-stage reinforcement learning method where prior to the DRL learning, we employ extreme learning machines (ELM) to initialize the parameters of actor and critic neural networks to a near optimal solution in order to significantly improve the learning performance. Finally, we illustrate our method in some typical path planning tasks. The experimental results have shown that our DRL method can converge much easier and faster. The resulting action deep neural network is able to successfully guide robots from any starting point in the environment to the goal position while following the optimal path and avoiding collision with obstacles.”
(2) In the introduction and related works sections, this paper focuses on deep reinforcement learning rather than on optimal path planning, not only the motivation of this study, but also the related works.
Answer: In this paper, we employ the DRL technique to produce the optimal action policy for mobile robots. We propose and focus on a novel deep reinforcement learning (DRL) approach for optimal path planning for mobile robots. Conventional optimal path planning methods often produce open-loop optimal control sequences instead of the closed-loop state feedback control policy for the discrete-time robot system. However, the closed-loop optimal action/control policy is better than open-loop optimal control sequences due to its robustness to the disturbances or noise in the real-world environment. The major goal of our proposed DRL method is to produce a closed-loop optimal action policy, and this study aims to overcome some major existing challenges such as slow learning process and low training data quality/efficiency in conventional DRL algorithms. However, conventional RRT, A*, and Bug approaches find only one open-loop optimal path from only one start position to the goal position, and need to re-plan a new path for a different start position or a disturbed middle position. In this revised version, we added more references of conventional path planning algorithms such as RRT, A*, Bug algorithms in Section 2. Related Work.
(3) The representations of the two algorithms do not agree.
Answer: We have revised Algorithm 3 and Algorithm 2 with more detailed explanations.
(4) Some equations are unlabeled.
Answer: We have labelled all the equations in this revised version.
(5) In the environment section, the proposed method in this paper are validated. However, this experiment lacks quantitative analysis and comparison with other excellent algorithms.
Answer: Thanks for the comment. We have added more experiments, even with the same training experience data, the conventional DDPG algorithm often leads to collision with obstacles in our experimental scenario. We will conduct a separate comparative study in the future due to its importance and the extensive amount of the work involved.
Reviewer 2 Report
The paper proposes a novel optimal path planning method based on dynamic programming (DP)-based data collection and deep reinforcement learning. After carefully reading the paper,I think it can be accpted in terms of the contributions. However,I have the following comments:1、There are some misspellings of symbols,such as the ?(?37,???) in the 263 line.
2、Is it possible to add some quantitative metrics or results that indicate the path is optimal?
Author Response
Thank you very much for all the comments. We highly appreciate your time and efforts to improve our paper. We have revised the paper based on your valuable feedbacks and we also address your comments here in this letter.
Reviewer 2:
The paper proposes a novel optimal path planning method based on dynamic programming (DP)-based data collection and deep reinforcement learning. After carefully reading the paper, I think it can be accepted in terms of the contributions. However, I have the following comments:
1、There are some misspellings of symbols, such as the (?37,) in the 263 line.
Answer: We have revised the paper. For example, we have replaced with to represent the shortest distance from point to the goal position, in Equation (4a).
2、Is it possible to add some quantitative metrics or results that indicate the path is optimal?
Answer: We have added more experiments to compare our proposed approach with the conventional DDPG algorithm. Even with the same training experience data, the conventional DDPG algorithm often leads to collision with obstacles in our experimental scenario while our proposed approach can rapidly converge to the optimal policy and paths.
Reviewer 3 Report
This paper presented a trajectory planning method based on deep reinforcement learning. However, the novelty of this paper is not clear. The structure also should be improved.
Some comments:
1. The structure of this paper must be improved. For example, the abstract is separated in several parts and indexed, which is not standard in my opinion.
2. The novelty of this paper is not clear. It seems that authors simply combined ELM and DNN. The reason for applying these methods is also not explained.
3. The simulation results are too simple to present the advantage of this method. The figures are also too rough.
Author Response
Thank you very much for all the comments. We highly appreciate your time and efforts to improve our paper. We have revised the paper based on your valuable feedbacks and we also address your comments here in this letter.
Reviewer 3:
This paper presented a trajectory planning method based on deep reinforcement learning. However, the novelty of this paper is not clear. The structure also should be improved.
Some comments:
- The structure of this paper must be improved. For example, the abstract is separated in several parts and indexed, which is not standard in my opinion.
Answer: We have revised the abstract. We have copied the revised abstract below for your convenience.
“In this paper, we propose a novel deep reinforcement learning (DRL) method for optimal path planning for mobile robots using dynamic programming (DP)-based data collection. The pro-posed method can overcome slow learning process and improve training data quality inherently in DRL algorithms. The main idea of our approach is as follows. First, we map the dynamic programming method to typical optimal path planning problems for mobile robots and present a new, efficient DP-based method to find the exact, analytical, optimal solution for the path planning problem. Then, we use high quality training data gathered using the DP method for DRL learning, which greatly improves training data quality and learning efficiency. Next we propose a two-stage reinforcement learning method where prior to the DRL learning, we employ extreme learning machines (ELM) to initialize the parameters of actor and critic neural networks to a near optimal solution in order to significantly improve the learning performance. Finally, we illustrate our method in some typical path planning tasks. The experimental results have shown that our DRL method can converge much easier and faster. The resulting action deep neural network is able to successfully guide robots from any starting point in the environment to the goal position while following the optimal path and avoiding collision with obstacles.”
- The novelty of this paper is not clear. It seems that authors simply combined ELM and DNN. The reason for applying these methods is also not explained.
Answer: We have revised the paper to address this comment. Please see Sections 1, 2 & 3 for details in this revised version.
Our proposed approach produces a closed-loop optimal continuous action/control policy via deep reinforcement learning, which is able to guide the mobile robot to move along the optimal path from any start position in the continuous free workspace to the goal position. However, conventional RRT, A*, and Bug approaches find only one open-loop optimal path from only one start position to the goal position, and need to re-plan a new path for a different start position or a disturbed middle position.
Conventional DRL approaches still have some critical challenges such as low quality training data, slow learning process or even often failure to converge, and local minimum issue. To overcome these challenges, our proposed approach is novel in the following three aspects: (1) the DP-based technique produces high-quality, most useful training data, (2) employing optimal initial training data and ELM learning to produce a near-optimal initialization for action policy and critic neural networks to significantly speed up learning process and overcome the local minimum problem, and (3) our modified DDPG algorithm with more samples in complicated regions to rapidly learn highly complex nonlinear optimal policy in these regions.
Specifically, DP-based training data contain the global optimal information. The derived optimal experience data include not only local information but also global and most useful information, and can effectively guide the learning process. Therefore, the learning process can efficiently learn from these best experience data and converge fast. For example, global optimal moving directions (i.e. actions) are the best information from the environment, contain much more useful information for learning than randomly collected low-quality experience data.
The major challenge of complex continuous policy representation using the neural network is prone to fall in a local minimum. ELM can effectively deal with this problem. ELM is one type of supervised learning, and is much more efficient to learn than conventional reinforcement learning algorithms since the target information is generally not available in conventional deep reinforcement learning approaches. The reason is as follows.
- DP-based optimal training data provide global optimal moving directions or actions and can be used as the optimal learning target.
- Given the input-to-hidden parameters of the actor and critic neural networks in DRL, ELM formulates the learning problem as a quadratic optimization problem, which has the closed-form solution resulting in rapid non-iterative learning, and more importantly, ELM can achieve the global optimal solution, which effectively reduce or even eliminate the local minimum problem inherent in DNN learning. Therefore, ELM provides a very good initial solution close to the global optimal solution in our experimental scenarios.
In this study, we propose a novel two-stage efficient DRL algorithm combining ELM and the modified DDPG (MDDPG). We employ both local experience data and global optimal experiences to guide the learning process to make the learning process converge to the optimal solution while avoiding the collision with obstacles using more local experiences in complicated regions such as the regions close to obstacles and ridges. In the free workspace far away from obstacles or ridge regions, the optimal policy is relatively smoother and requires a smaller amount of training samples. Our rapid MDDPG learning benefits from the following three aspects:
- near-optimal initialization via fast ELM learning, which makes MDDPG learning process require a smaller number of training iterations,
- global optimal experiences, the most useful training data to guide effective learning of optimal policy, and
- more local experiences in complicated regions (a) close to obstacles for fast learning of collision avoidance since the optimal policy changes significantly in the regions close to obstacles, (b) in ridge regions to fine-tune more complex nonlinear relationship/function of optimal policy a(s) between action a and state s since different optimal moving directions diverges in ridge areas.
- The simulation results are too simple to present the advantage of this method. The figures are also too rough.
Answer: We have re-produced some figures with high resolutions. We also add more experiments to compare our proposed approach with the conventional DDPG algorithm. Even with the same training experience data, the conventional DDPG algorithm often leads to collision with obstacles in our experimental scenario while our proposed approach can rapidly converge to the optimal policy. Please see Section 4. Experimental Results for details.
Reviewer 4 Report
In this paper, the authors used Deep reinforcement learning to construct an optimal path for a mobile robot. In order to speed up the reinforcement learning process and improve the convergence performance; we also proposed a two-stage deep reinforcement learning method, in which ELM is employed for weight parameters initialization of actor and critic networks. This paper is conceptually good, but there are some comments on this paper to make the paper improved.
1. The time complexity of the algorithm compared using the existing ones.
2. There are several algorithms for robot path planning through learning such as RRT, A*, Bug algorithm, etc. However, the authors are not considered them in the literature.
3. The proposed work must compare with the similar approaches such as RRT, A*, Bug2 algorithm to show the superiority.
4. The reasons for achieving superiority of the proposed work over the existing ones is not clearly mentioned in this paper.
5. The limitations of the proposed work is not discussed in the paper.
6. The use of experience reply buffer is not found in the paper.
7. The results are not showing that the proposed work is optimal or convergence quickly to reach the solution. More prominent results to be provided.
8. The introduction is not clear. The authors must include the need of this work in the current scenarios, and also a motivation example.
9. Some of the figures are not clear. the authors must provide a clear and good resolution images.
Author Response
Thank you very much for all the comments. We highly appreciate your time and efforts to improve our paper. We have revised the paper based on your valuable feedbacks and we also address your comments here in this letter.
Reviewer 4:
In this paper, the authors used Deep reinforcement learning to construct an optimal path for a mobile robot. In order to speed up the reinforcement learning process and improve the convergence performance; we also proposed a two-stage deep reinforcement learning method, in which ELM is employed for weight parameters initialization of actor and critic networks. This paper is conceptually good, but there are some comments on this paper to make the paper improved.
- The time complexity of the algorithm compared using the existing ones.
Answer: Our proposed DRL algorithm integrates three major different techniques: the DP-based technique to collect optimal training data, fast ELM learning for parameter initialization, and the modified DDPG algorithm to significantly improve the learning performance which is an iterative learning process, and the learning performance does not depend only on the number of training samples, but also depend on other factors such as training data quality. Our algorithm is significantly different from conventional optimal path planning algorithms such as RRT, A* and Bug algorithms, and it is a challenging task to derive the exact time complexity of our proposed algorithm.
- There are several algorithms for robot path planning through learning such as RRT, A*, Bug algorithm, etc. However, the authors are not considered them in the literature.
Answer: We have added RRT, A*, Bug algorithms in Section 2. Related Work in this revised version.
- The proposed work must compare with the similar approaches such as RRT, A*, Bug2 algorithm to show the superiority.
Answer: This is a good point, we plan to conduct a separate comparative study in our future research due to its importance and the extensive amount of the work involved.
The RRT, A*, and Bug approaches aim to find one optimal path from only one start position to the goal position, i.e. the open-loop sequences of points along the trajectory. When the start position changes or the mobile robot deviates from the optimal path caused by disturbances, we have to redo the path planning again from the new start position. However, our proposed method produces a closed-loop optimal action/control policy, once we learn the continuous optimal policy via DRL, for any start position or any middle disturbed position, the robot can always move along the optimal path starting from any new position in the continuous free workspace based on the optimal policy without re-planning. Therefore the purpose of our proposed approach is different from RRT, A*, and Bug approaches, and is better.
- The reasons for achieving superiority of the proposed work over the existing ones is not clearly mentioned in this paper.
Answer: Based on this comment, we have revised the paper. Our proposed approach produces a closed-loop optimal continuous action/control policy via deep reinforcement learning, which is able to guide the mobile robot to move along the optimal path from any start position or any middle disturbed position in the free workspace. However, conventional RRT, A*, and Bug approaches find only one optimal path from one start position to the goal position, and need to re-plan the new path for the different start position.
Conventional DRL approaches still have some critical challenges such as low quality training data, slow learning process or failure to converge, and local minimum issue. To overcome these challenges, we propose a novel efficient DRL learning approach that significantly improve the learning performance mainly because of the following three reasons: (1) the DP-based technique provides high-quality, most useful training data for deep reinforcement learning, (2) employing DP-based optimal initial training data and fast ELM learning to produce a near-optimal initialization for action policy and critic networks to significantly speed up learning process and overcome local minimum problem, and (3) the modified DDPG algorithm with more training samples of global optimal data and randomly collected data in complicated regions to rapidly learn highly complex nonlinear optimal policy in these areas.
- The limitations of the proposed work is not discussed in the paper.
Answer: In the current form, the proposed work cannot handle more complicated scenarios such as obstacles of arbitrary shapes and multiple mobile robots. However, we plan to investigate these more complex scenarios in the near future as we discuss in Section 5. Conclusion and Future Work.
- The use of experience reply buffer is not found in the paper.
Answer: The use of experience replay buffer is used as “replay buffer R” in Algorithm 3 in Section 3.2.5 Two-Stage DRL Algorithm.
- The results are not showing that the proposed work is optimal or convergence quickly to reach the solution. More prominent results to be provided.
Answer: In theory, DRL can produce the optimal policy in the probability of one with enough time and enough training data. But in most real-world applications, we have limited time and resources. We have added more experiments to show that the conventional DDPG algorithm usually fail to get a feasible solution even with same training experience data, that is, the learned policy leads to collision with the obstacles in the experimental scenarios.
- The introduction is not clear. The authors must include the need of this work in the current scenarios, and also a motivation example.
Answer: Based on this comment, we have revised the introduction and other sections in this revised version.
By using DRL in this study, we aim to learn the optimal action policy, i.e. closed-loop feedback for real-time navigation of mobile robots in a 2D environment. Our optimal closed-loop action policy, which is for any starting point and cover all trajectories in the continuous free workspace, and is better than the optimal open-loop action sequence which is only for one start point and is therefore one trajectory. The optimal action policy allows real-time fast optimal response as the robot moves around in a complex environment. Closed-loop action policy overcomes the shortcomings of open loop action sequences. It is also robust to the disturbances or noise and reduces the deviation caused by disturbances since effects of the disturbances are automatically compensated for. For example, in the case of the robot’s state deviating from the optimal path caused by disturbance, the closed-loop action policy can mitigate the deviation from the original trajectory without accumulative errors. The optimal policy can guarantee that the robot can still move along a new optimal path starting from the new disturbed state, which will be close to the original optimal trajectory while the open-loop action sequence maybe make the robot move far away from the original optimal trajectory due to accumulative errors. Therefore, the closed-loop optimal policy is crucial for the real-time optimal navigation for mobile robots. Although in theory DRL has the potential to achieve superhuman performance, in practice, it is very challenging to efficiently learn the parameters of the optimal policy i.e. the continuous actor neural network for specific applications using DRL. In conventional DRL algorithms, random experience data or experience data collection using ε-greedy strategy cannot provide high-quality experience training data with very useful information in general since these low-quality experience data usually are not optimal experiences. Therefore, this leads to a slow long learning process with these low-quality training data in general for conventional DRL algorithms to achieve the optimal action policy.
Conventional DRL approaches still have some critical challenges such as low quality training data, slow learning process or failure to converge, and local minimum issue. To overcome these challenges, our proposed approach is novel in the following three aspects: (1) the DP-based technique provides high-quality, most useful training data for deep reinforcement learning, (2) employing DP-based optimal initial training data and fast ELM learning to produce a near-optimal initialization for action policy and critic networks to significantly speed up learning process and overcome local minimum problem, and (3) the modified DDPG algorithm with more training samples of global optimal data and randomly collected data in complicated regions to rapidly learn highly complex nonlinear optimal policy in these areas.
- Some of the figures are not clear. the authors must provide a clear and good resolution images.
Answer: We have provided good resolution images for some of the figures in this revised version.
Round 2
Reviewer 1 Report
I have no other suggestions.
Author Response
We sincerely appreciate all your valuable comments to improve our paper!
Reviewer 3 Report
This paper can be accpeted.
Author Response

(The authors gave the same response as above.)

Reviewer 4 Report
The authors made lots of efforts to revise the paper and it is well improved over the previous version. However, few corrections made before publishing it.
1. Some of the figures are not clear, and it is very difficult to understand the text.
2. The Algorithm 1, 2 and 3 are written in pseudo code format.
3. Its not good to start a sentence with 'And' in research papers.
4. Computational efficiency of the proposed method is discussed and compared with the existing ones.
Author Response
Thank you very much for your valuable comments and suggestions! We greatly appreciate your efforts and time to improve our paper. We have revised our paper according to your feedbacks and we also addressed all your comments here in this letter.
Reviewer 4.
Comments and Suggestions for Authors
The authors made lots of efforts to revise the paper and it is well improved over the previous version. However, few corrections made before publishing it.
- Some of the figures are not clear, and it is very difficult to understand the text.
Answer: We have re-produced Figures 8 and 9. We have also added more detailed explanation in the figure caption and text. Please see Section 4.2 for details.
- The Algorithm 1, 2 and 3 are written in pseudo code format.
Answer: Based on the comment, we have revised Algorithms 1, 2 and 3 in this revised version.
- Its not good to start a sentence with 'And' in research papers.
Answer: Based on the comment, we have made corrections in this revised version.
- Computational efficiency of the proposed method is discussed and compared with the existing ones.
Answer: We have conducted the experiment to compare our proposed method with the existing original DDPG algorithm in Section 4.4.2 and made some revision. In terms of the computational efficiency, our proposed method used only 70,000 training iterations to converge to the optimal solution while the original DDPG algorithm used 250,000 training iterations but still failed to converge to a feasible solution. This result demonstrates that our method is much more efficient than the original DDPG algorithm. Please see the last paragraph and Figs. 17 & 18 in Section 4.4.2 for details.